# Molecular Dynamics Simulation of Sintering Densification of Multi-Scale Silver Layer

**DOI:** 10.3390/ma15062232

**Published:** 2022-03-17

**Authors:** Peijie Liang, Zhiliang Pan, Liang Tang, Guoqi Zhang, Daoguo Yang, Siliang He, Haidong Yan

**Affiliations:** 1Guangxi Key Laboratory of Manufacturing System & Advanced Manufacturing Technology, School of Mechanical and Electrical Engineering, Guilin University of Electronic Technology, Guilin 541004, China; liangpj2019@163.com (P.L.); zpan@guet.edu.cn (Z.P.); tangliang@guet.edu.cn (L.T.); g.q.zhang@tudelft.nl (G.Z.); daoguo_yang@163.com (D.Y.); 2Delft Institute of Microsystems and Nanoelectronics (Dimes), Delft University of Technology, Mekelweg 6, 2628 CD Delft, The Netherlands; 3Institute of Semiconductors, Guangdong Academy of Sciences, Guangzhou 510650, China; 4Hang Zhou Global Scientific and Technological Innovation Center, Zhejiang University, Hangzhou 311200, China

**Keywords:** molecular dynamics, initial coating morphology, random algorithm, semi in-situ observation

## Abstract

Based on molecular dynamics (MD), in this study, a model was established to simulate the initial coating morphology of silver paste by using a random algorithm, and the effects of different sizes of particles on sintering porosity were also analyzed. The MD result reveals that compared with the sintering process using large-scale silver particles, the sintering process using multi-scale silver particles would enhance the densification under the same sintering conditions, which authenticates the feasibility of adding small silver particles to large-scale silver particles in theory. In addition, to further verify the feasibility of the multi-scale sintering, a semi in-situ observation was prepared for a sintering experiment using micro-nano multi-scale silver paste. The feasibility of multi-scale silver sintering is proved by theoretical and experimental means, which can provide a meaningful reference for optimizing the sintering process and the preparation of silver paste for die-attach in powering electronics industry. In addition, it is hoped that better progress can be made on this basis in the future.

## 1. Introduction

Sintering using silver particles is considered to be one of the ideal bonding techniques for the wide bandgap semiconductors, e.g., silicon carbide (SiC) and gallium nitride (GaN) devices, because this technique is capble of forming a high-melting-point sintered silver joint with sintering temperatures lower than 300 ℃ [1,2,3,4].

Nanosilver paste is considered an ideal bonding material since it can sinter at a temperature lower than 250 ℃ without pressure and operate at a temperature higher than 400 ℃ [5,6,7,8]. However, the difficulty of preparation and preservation of nanosilver paste results in a high cost in the sintering process. With the continuous improvement of power performance requirements of power electronic equipment, realizing the maximum commercial value of sintered silver is no longer limited to optimizing the sintering process, sintered joint performance, and energy consumption. Rather, optimizing the formula of silver paste is also very important There has been a rapid increase in the demand for power electronics devices because of the strong growth of electric vehicles. In order to maximize commercial value, while considering the sintering behaviors, mechanical properties of joint, energy consumption, and safety of silver particles, researchers are focusing on optimizing the silver particle formula in silver particles paste to achieve compromise solutions. Figure 1a shows the scanning electron microscope (SEM) image of silver nanoparticles after sintering without pressure. Sintering neckings formed between the nano particles can be clearly observed due to the easy diffusion between the adjacent particles [9]. However, the rapid diffusion of silver nanoparticles formed a sintered layer with high porosity during the sintering process. Figure 1b shows the SEM image of silver micron particles sintered under the same sintering conditions as Figure 1a. Compared with Figure 1a, the necking between micron particles is few and inconspicuous. It is difficult to form diffusion necks in the pressureless sintering process due to the low surface energy of large particles. Therefore, applying 10–30 MPa or even higher assist-pressure during the sintering process was used to obtain a reliable micron silver sintered joint [10,11].

Figure 1a,b showed the SEM of different particles after pressureless sintering at 250 ℃ for 60 min by Suganuma’s team from Osaka University. It demonstrates the advantages and disadvantages of different scale granular silver sintered solder joints, which are verified by using the experimental method of controlling variables. Based on this study, a pressureless sintering process was proposed by adding the silver nano particles in the silver micro paste to improve the sintering driving force and reduce the sintering temperature, as shown in Figure 1c,d [12]. In our previous study, we were surprised to find that, after pressureless sintering at 250 ℃ for 60 min, a particle with a diameter of 3.6 μm can form a sintering necking with a silver particle with a diameter of 27.6 μm, as shown in Figure 1e. These results prove the feasibility of low cost nano-micron multi-scale silver paste which prepared by appropriate addition of nanoparticles in low-cost micron silver paste as the potential bonding materials for sintering. The reliability of multi-scale silver sintering as a bonding layer has been clarified in many previous research works [12,13,14,15]. However, in past demonstrations of multi-scale silver sintering, most of the experiments are based on the densification analysis at the macro level, and the densification analysis at the micro molecular level is not explained. The initial dynamic diffusion evolution of sintering using multi-scale silver particles, which is important to reveal the sintering mechanism, is not clear and has thus far been difficult to observe. In order to further reveal the micro mechanism of mixed scale particle sintering, we use molecular dynamics simulation to reveal the difference of diffusion mechanism between mixed particles and large particles.

MD is a multidisciplinary technology that relies on Newtonian mechanics and integrates physics, chemistry, and mathematics to simulate the molecular system’s motion, which is also widely used in studying the diffusion mechanism of silver sintering [16]. In this study to simulate more authenticity, the sintering of different sizes of particles for MD analysis was prepared with a random algorithm. It is interesting to note that using the random algorithm could well simulate the randomness of particles stacking and arrangement in the silver paste layer, which will be described in the next chapter. After simulation, to further verify the feasibility of the multi-scale sintering, a semi in-situ observation was prepared for a sintering experiment using micro-nano multi-scale silver paste.

## 2. Method

### 2.1. Methodology of Simulation

#### 2.1.1. Molecular Dynamics Interatomic Potential

Sintering technology forms a dense joint based on the particles’ diffusion below the melting temperature, according to the embedded atom method (EAM) [17]. Using the Large-Scale Atomic Molecular Massively Parallel Simulator (LAMMPS), this study combined the EAM to Simulate the diffusion between particles with different scales at different temperatures. The simulation atom-style was selected as atomic because sintering bonding is a diffusion process between particles. The timestep unit is related to the selected atom-style; In this study, the timestep size is set to the default values 0.001 ps to steadily solve Newton’s equation of motion.

The molecular dynamics model is composed of atoms arranged in a specific order, and it could cause geometric distortion if the simulation’s model size is too small. In order to reduce the model distortion, we had established the minimum particle radius of 10 Å in this study. The three style particles with radii of 10 Å, 30 Å, and 50 Å were designed by the “equal interval” method. As in past studies [18,19], the model’s crystal orientation could affect the simulation results. To make the simulation more closely fit the actual situation, the particles in the simulation box as described in the next section were set at different crystal orientations by random placement.

#### 2.1.2. Random Algorithm

Matlab is a matrix-based language that can provide a most natural expression of computational mathematics and provides researchers with the required algorithm programming for designing and analyzing the data [20].

In order to simulate a random mix like the particles of the silver paste coating on the substrate in the real application, Matlab used mathematical analysis and computation to design a simulation box with the random algorithm. The circles with different radius are randomly arranged in proportion according to the random function in the box. However, it is difficult to avoid intersecting and overlapping circles because generating random circles generates random dots and draws circles. Thus, in this step, we design the circles with a proposed ratio by the random function and remove the circles which intersect or overlap, and, finally, output the random circle’s position and size which had been retained.

In this study, we had designed a Half-two-dimensional (H2D) box of 500 × 6 × 300 Å along *X*, *Y*, and *Z* directions with the periodic boundary to simulation the densification process of one cross-section of the silver layer to reduce simulation redundancy, shorten the running time, and achieve a more precise understanding of the densification process In order to understand the effect of multiscale on densification, the following three groups of different mass ratio groups are set in this study as shown in Table 1.

However, the mass ratio in the grouping must transform into the corresponding particle ratio before modeling. In the H2D simulation in this study, the conversion of mass ratio and particle ratio is associated with cross-sectional particle area, and the around ratios of the corresponding particles are shown in Table 2.

The particle ratio was substituted into the random equation in MatLab, and we obtained the random arrangement of various circles shown in Figure 2. Position coordinates of the circle generated in the random function are extracted and substituted into the MD model. In order to calculate the porosity easily and make the simulation the silver paste coating on the substrate, we designed the upper and lower silver plates in the *Z* direction.

### 2.2. Semi In-Situ Observation Sintering of Multi-Scale Silver

In-situ sintering observation is a method to observe the diffusion bonding process of silver particles by scanning electron microscope without resampling during the sintering, which observes the morphology change of the particles better than the traditional subsection sampling method [21,22].

In this study, we used a scanning electron microscope equipped with a protochips single tilt heating holder with a maximum heating temperature of 1200 ℃ to observe the multi-scale silver paste sintering semi in-situ.

During the sintering, in order to avoid electron beam radiation, only open the gun valve when sampling the sample [23]. We referred to the micron silver sintering profile by K. S. Siow [24], and we set the sintering temperature to 250 °C for 10 min on the pressureless setting.

## 3. Result and Discussion

### 3.1. Sintering Densification of Silver Particles with Different Proportions

Figure 3 shows the morphology of the sintering densification process of groups A, B, and C sintering at 300 °C. Comparing the sintering processes of A and B, it is obvious to find that group B with small particles is better than group A in the reduction of porosity, the reduction of which is more significant for group C with smaller silver particles based on group B. In the initial stage, driven by thermal energy, the particles in the sintering layer begin to “move” and “attract” the particles around them, and begin to gather to prepare for the start of sintering; which also called the first stage of sintering. With the continuous sintering, the grain boundary energy of particles begins to weaken and the particles begin to diffuse each other, which is a longer process (10–1000 ps). However, with the subsequent sintering, the change of sintering pore decrease becomes weaker, which is more clearly reflected in the change curve of porosity of groups in Figure 4. Multiscale group C not only shows lower porosity than group A and B after sintering but also reduces the area of pore due to its complementarity in the gap of large particles. In the actual production process, the large pore will lead to uneven heat dissipation, resulting in heat concentration and local thermal resistance increasing, which will also affect the reliability of packaging.

Figure 4a shows the changing profiles of different particle ratios’ porosity and Figure 4b shows the change rate of porosity. The initial stage is the early stage of sintering, so this stage has the maximum driving force in the whole pressureless sintering process. It shows that the porosity decreases rapidly which is shown in Figure 4a, and the diffusion rate is the fastest in the whole sintering, as shown in Figure 4b. With the continuous sintering, the porosity changes gradually from the initial sharp change to stability. We also found that the larger the particle, the earlier it tends to be stable; see Figure 4b. It is also found from Figure 3 that the larger pores which are found at 500 ps hardly change after 1000 ps, while the smaller pores will be eliminated or further reduced. This is because the bonding completed early greatly weakens the surface energy in the system, which leads to the application temperature being insufficient to provide sufficient diffusion driving force. Porosity slight shrinkage leads to the diffusion rate slowing down.

Combined with the curves from Figure 4, we intuitively find that the porosity and porosity change rate of the simulation group with small particles is better than without the small particles. The relative surface energy is inversely proportional to the particle size, and adding small particles to large particles can improve the sintering driving force. The relationship between particle size and corresponding drive force follows [16]: (1)$$\sigma =\gamma (\frac{1}{R_{1}}+\frac{1}{R_{2}})$$
where $\sigma_{i}$ is the sintering driving force, $\sigma_{i}$ is the material surface energy, and $R_{1}$ and $R_{2}$ are the radii of two particles that diffuse each other. It can be seen from the graph in Figure 4b that the maximum sintering diffusion change rate in the initial stage of mixed particle sintering group C can reach 0.67, which is twice that of group A with only large particles reaching 0.32.

From the above results, with the same sintering temperature and time, multi-scale silver is better than large particle sintering in sintering porosity. Moreover, the small particles in multi-scale silver act as a “filler”, which also greatly reduces the average volume of pores. In order to further understand the diffusion mechanism of silver particles with different sizes in the sintering process, different silver particles’ change of necking was simulated and analyzed. This will be described in the next section.

### 3.2. Evolution of Sintering Silver Neck Size

As shown in Figure 5 the bonding of the following three different size particles pair was studied. The particles pair in the model set different crystal orientations to avoid the contingency of the same crystal orientation. In this chapter, we studied the necking of different particles’ radii. We cut 474, 13,240 and 61,370 silver atoms from a large silver FCC supercell to form silver particle pairs with radii of 10 Å, 30 Å and 50 Å respectively as shown in Figure 5, the initial distance of 2 Å between particle pair. As shown in Figure 4, the porosity in the sintering process decreases rapidly within the first 200 ps, and then the densification slows down. Therefore, in this chapter, the simulation mainly takes the first 300 ps for study.

Figure 6 shows the simulated value and the simulated value’s fitting curve of x/r (*x* is the radius of diffusion neck and *r* in the radius diffusion particle.) of different particles’ radii during the sintering. As shown in Figure 6, the necking of particles with radius of 10 Å grows and diffuses rapidly, and the x/r reaches 1 at about 8 ps (the necking was completed). Figure 7 shows the profile of total potential energy during the necking of particles with a radius of 10 Å. Two particles approach and begin to diffuse under the driving force provided by thermal energy. The relative potential energy of the system decreases to provide diffusion kinetic energy. When the diffusion necking is completed, the system’s kinetic energy begins to be constant, the relative potential energy stops decreasing and remains stable. As the thermal energy in the system continues to be maintained, some atoms will cross the potential barrier and further diffuse, and potential energy will also change, as shown in the potential energy fluctuation at 250 ps in Figure 7.

However, under the same simulation conditions, the growth rate of particles with radii of 30 Å and 50 Å is slower, and the x/r with the radius of 30 Å tends to be stable when it increases to about 0.7, and the radius of 50 Å tends to be stable when it increases to about 0.52. This difference further indicates that small particles can diffuse more easily than large ones at the same temperature This difference further indicated that small particles are more easily diffused than large particles at the same temperature. The process of necking size accords with the diffusion formula [25]:(2)$${\left(\frac{x}{r}\right)}^{n}=\frac{Bt}{{\left(2r\right)}^{m}}$$
where *t* is the sintering time, *n* and *m* are constants dependent on the specific transport mechanism, and *B* is a term made of material and geometric constants. Combined with Equation (Equation 2) and the trend of measured data in Figure 6, the radii of 30 Å and 50 Å will continue to increase until the maximum values are reached if the simulation time or sintering temperature increases.

Small silver particles were doped into large silver particles to shorten the sintering bonding time and increase the sintering neck of large particles in this study, as shown in Figure 8. Figure 8a shows the sectional of 50 Å particles dope with 10 Å particles. In order to observe the diffusion process, the particles were divided into different groups and colored, respectively. As driven by thermal energy, the small particles approach two large particles and diffuses at 1 ps. With the sintering continuing, small particles diffuse into and fill the bonding gap of two large particles, and the diameter of the bonding neck reached 67.8 Å at 10 ps. However, under the same conditions, the diffusion diameter of only large particles is only 28.69 Å. With continued sintering, small particles continue to “climb” along with two large particles, further forming bonds with large particles. At the end of sintering, the neck of the sintering group with small particles shrinks to 66.46 Å, which is larger than 53.66 Å sintering with only large pair particles.

Small particles can “close” two large particles far away to form necking to increase the density of sintering, as shown in Figure 9. The initial distance of the two large particles in Figure 9a,b is 10 Å and the difference between them is that Figure 9a had added some small particles in the middle and the lower part between two large particles. Figure 9a shows that large particles pair get close together and form necking due to the pulling of small particles’ diffusion at the bottom. This is because small particles provide a certain pulling force for large particles to bond. Driven by this pulling force, the particles can close together and form necking. This can also be confirmed from the particle pair in Figure 9b which, without small particles, always keep the distance during the simulation process.

From the above results, whether from the simulation results of silver layer sintering or silver neck diffusion analysis, the evidence points to the superior characteristics of multi-scale silver sintering. From the perspective of molecular dynamics, the small silver particles of multi-scale silver do have a good densification effect on the sintering of large particles. However, this micro analysis does lack some support in practical application. Considering this, in order to further verify this feasibility, we will carry out semi in-situ sintering of multi-scale silver sintering in the next chapter.

### 3.3. Result of Semi In-Situ Observation

From Figure 10, the diffusion bonding process of Ag particles during the sintering by semi in-situ SEM is clearly shown. From the in-situ sintering diagram, particle pair B which is closed in the initial stage begins to form an obvious diffusion bonding neck when the sintering is carried out for 6 mi, as shown in Figure 10b. In the subsequent sintering, the diffusion silver neck of the B particle pair continues to grow. At the same time, nanosilver particles around the particle pair continue to diffuse to the large particles, also resulting in the also growth of the large particle pair while bonding. The bonding reaction between micron particles and nanoparticles in the A particle pair increase the feasibility of multi-scale sintered silver. For the A particle pair, we can see that the two particles are not in contact before sintering. However, after 8 min sintering, the particle pairs are driven by the nanoparticles and form a necking connection. Also as the particle pair which initial distance was 2.06 μm is shows in Figure 10a, and shorten to 0.6 μm after sintered 8 min. At the same time, the radius of the particle pair also increases from the initial 0.53 μm to 1.14 μm. This sintering change is consistent with the simulation results in Figure 9, which shows that the small silver particles of multi-scale silver paste can provide good sintering driving force in large particles, which can provide better density.

In this study, the micro silver paste with nano-silver is sintered for 8 min without pressure to form a densification bonding, which is much shorter than the traditional pure micro silver paste. Multiscale silver sintering is shortened not only the sintering time but also reduces the sintering temperature, which is also one of our future research directions.

## 4. Conclusions

Using molecular dynamics, combined with MatLab random arrangement,he sintering densification mechanism of silver with different mixed particle ratios has been studied, and the neck growth mechanism of a silver particle pair in the sintering process was analyzed. Combined with the validation experiment, the following conclusions can be summarized:With the same conditions such as sintering temperature and time, adding small silver particles to large silver particles is beneficial to reduce porosity.During the initial sintering stage, the x/r of the small silver particle pair is faster than the large silver particle pair due to higher surface diffusion under heating. Moreover, the x/r of a small particle pair is higher than that of a large particle pair at the same temperature due to the high surface diffusion energy. Using this mechanism improves the radius of the bonding neck and shortens the sintering time by adding small silver particles to large particles.In the multiscale silver sintering experiment, nano-silver as the medium of micron silver bonding could effectively shorten the sintering time and reduce the effective sintering temperature to improve the packaging yield.

## Figures and Tables

**Figure 1 materials-15-02232-f001:**
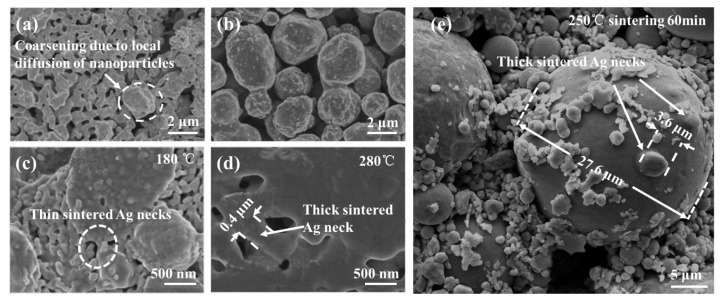
The SEM image of Ag sintered layer with (**a**) nano-silver and (**b**) micro-silver by Kazohiko Suganuma et al., adapted with permission from Ref. [9], 2022 Chuantong Chen; (**c**,**d**) are the sintering morphology of multi-scale silver at different sintering temperatures by H D. Yan et al., adapted with permission from Ref. [12]. 2022 Haidong Yan; (**e**) super large silver particles and micron silver particles formed silver neck after sintering.

**Figure 2 materials-15-02232-f002:**
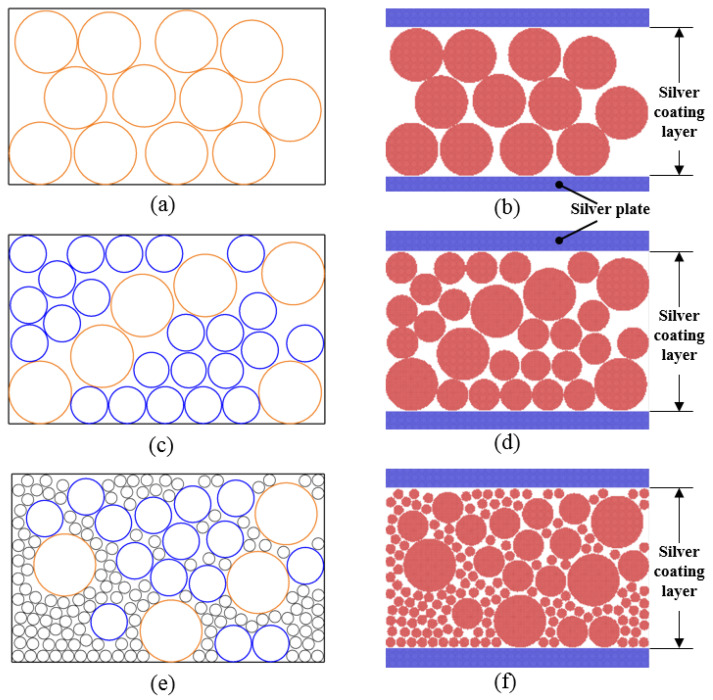
The circles’ random placement by MatLab with diameter ratio from Table 2 with (**a**) group A, (**c**) group B, and (**e**) group C. (**b**,**d**,**f**) were the MD model based on randomly placed data to (**a**,**c**,**e**), respectively.

**Figure 3 materials-15-02232-f003:**
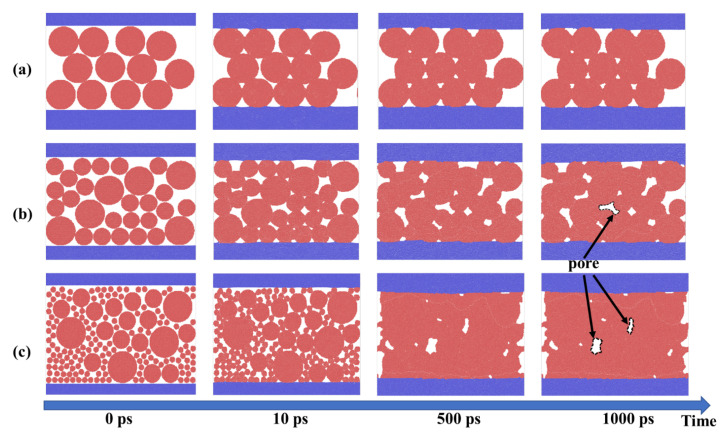
Morphology of sintering densification process of (**a**) group A, (**b**) B, and (**c**) C at 0 ps, 10 ps, 500 ps, and 1000 ps.

**Figure 4 materials-15-02232-f004:**
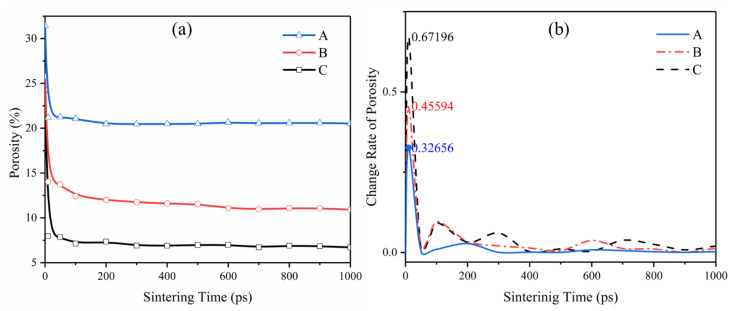
The profile of (**a**) porosity change with sintering time and corresponding (**b**) change rate of porosity.

**Figure 5 materials-15-02232-f005:**
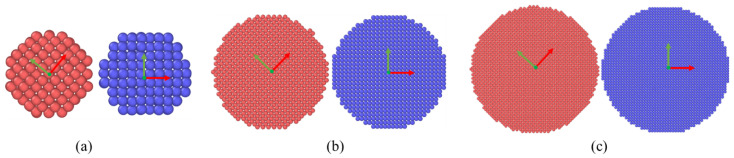
The Silver particles pair with different radius: (**a**) 10 Å, (**b**) 30 Å, and (**c**) 50 Å.

**Figure 6 materials-15-02232-f006:**
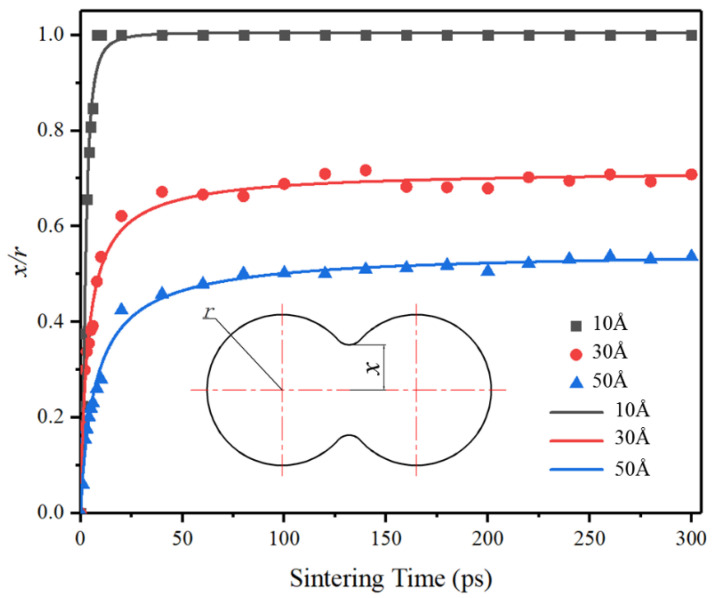
The measurement data and fitting curve of *x*/*r* with the time of silver particles pair with different radius sintered at 300 °C. The fitted curve conforms to the growth of the logarithmic function, which is consistent with the rapid change to slow the change of measured data.

**Figure 7 materials-15-02232-f007:**
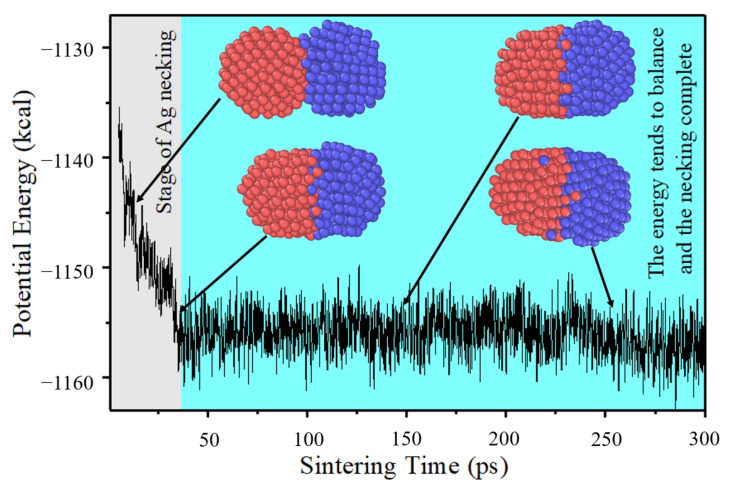
Potential energy with time during sintering of particle pair with a radius of 10 Å.

**Figure 8 materials-15-02232-f008:**
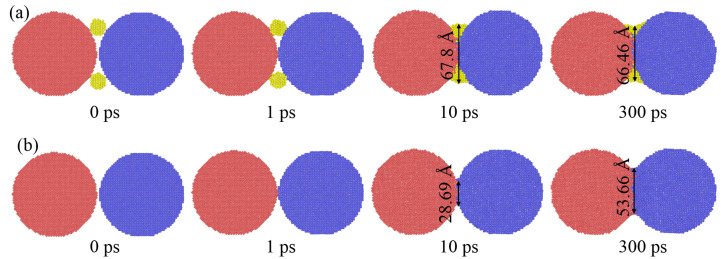
MD profile of particle pair with the radius of 50 Å; (**a**) doping the silver particles with radius of 10 Å, and (**b**) control group without any doping.

**Figure 9 materials-15-02232-f009:**
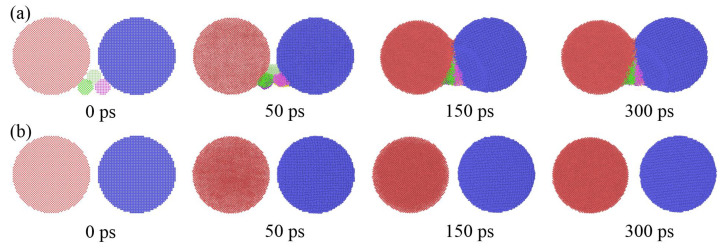
MD profile of a particle pair with radius 50 Å; (**a**) Add small particles in the middle and lower part between large particle pair and (**b**) control group without any doping.

**Figure 10 materials-15-02232-f010:**
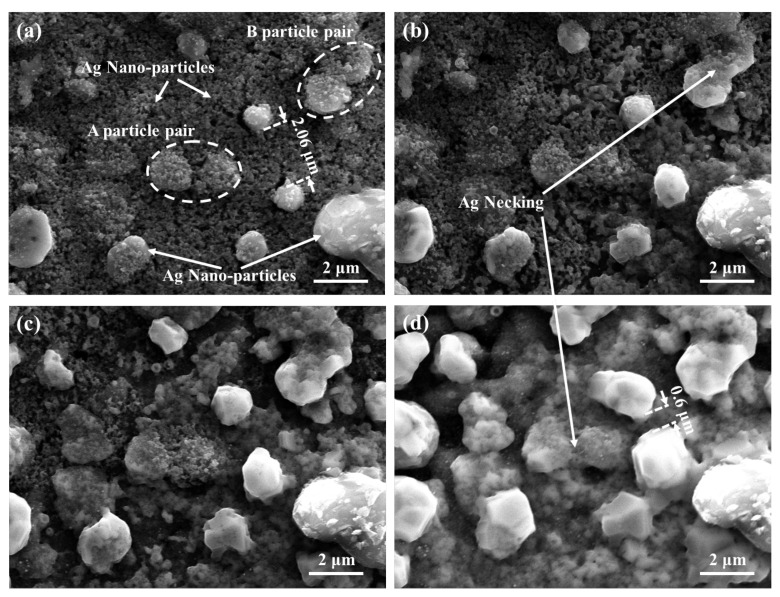
The SEM image of In-Situ multiscale silver sintering at different time steps (**a**) dried, (**b**) after sintering 6 min, (**c**) after sintering 7 min, and (**d**) after sintering 8 min.

**Table 1 materials-15-02232-t001:** Grouping with different mass ratios.

Group	10 Å	30 Å	50 Å
A	0	0	1
B	0	1	1
C	1	1	1

**Table 2 materials-15-02232-t002:** Particle ratio corresponding to different mass ratio (base on H2D).

Group	10 Å	30 Å	50 Å
A	0	0	1
B	0	3	1
C	25	3	1

## Data Availability

Not applicable.

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
