# Peer review of "Molecular Dynamics Simulation of Sintering Densification of Multi-Scale Silver Layer"

_materials, 2022, doi:10.3390/ma15062232_

Round 1

Reviewer 1 Report

The article describes about the modelling of sintering densification of different sized silver particles using molecular dynamics. The feasibility of multi-scale sintering has also been tested via semi in situ SEM studies. There are several critical aspects to be cleared before publishing in ‘Materials’.

* In the Molecular Dynamics model, whether the authors are reporting atoms or particles with radii of 10, 30 and 50 Å. It is confusing between atoms and particles. Has the model considered the number of atoms in a single silver nano and micro particle? 

Lines 165-167 - Particles with a radius of 10, 30 and 50 Å. It is recommended to explain how these could be considered as nano and micro-particles. It has been reported by Zhang et al (Computational Materials Science 125 (2016) 105–109​) that in order to prepare a nanoparticle with realistic atomistic structure, single silver nanoparticle with 6663 atoms is cut out from a large silver FCC supercell within a radius of 3.0 nm. In the present study, this needs to be elaborated.

* In Fig 1 a-b, SEM images of sintered Ag surface with nanoparticles and microparticles reported by (C. Chen, K. Suganuma, Materials and Design 162 (2019) 311–321) are shown. However, in the figure caption it is written as by Kazuhiko Sugiura et. al. This needs to be clear with details of proper permissions for reuse.

* It would be interesting to explain why the authors selected multi-scale silver particles with micro and nano sizes. Why the sub-micron size was excluded for the simulation?

* The porosity of the sintered morphology has been considered only on the surface. Considering the practical aspect, the effect of different size on porosity along the cross-section of sintered Ag paste should also be addressed as the porosity will be different on the surface and cross-section.

* Lines 45, 243, 244- particle diameter of 3.6 m, 27.6 m, 2.06 m, 0.6 m- these are serious errors 

* Line 46- feasibility of low cost nano-micron multi-scale silver particles as the potential bonding materials for sintering. It is suggested to explain how low cost can be achieved.

* Line 70 - vacancy formation energy - Embedded atom method has been successfully used for predicting phase transformations, surface energy, and vacancy formation energy in metal and alloy systems. In the present context what is the relevance of prediction of vacancy formation energy? ​

* How was the semi-in situ sintering monitoring conducted with the aid of SEM? This needs to be elaborated.

* Fig.10 - How was A particle pair and B particle pair discerned from the SEM image?

Author Response

We thank you for the positive comments and suggestions, which increased the scientific quality of this manuscript. We have revised our manuscript according to the comments, with the changes highlighted in the revised manuscript. Specific point-by-point responses to your comments are listed in the below. We believe that we have addressed the reviewer’s comments satisfactorily. We hope the revised manuscript is now suitable for publication and looking forward to hearing from you.

  1. In the Molecular Dynamics model, whether the authors are reporting atoms or particles with radii of 10, 30 and 50 Å. It is confusing between atoms and particles. Has the model considered the number of atoms in a single silver nano and micro particle?

Response:

Thank you for this comment. These are the radii of particles, about the modeling, In the simulation, considering the effective simulation time of molecular dynamics, our simulation modeling particles are within the Angstrom level ( 1Å=0.1nm ). The purpose is to demonstrate the densification mechanism of multi-scale silver sintering and the feasibility of multi-scale silver sintering through these simulations.

2.Lines 165-167 - Particles with a radius of 10, 30 and 50 Å. It is recommended to explain how these could be considered as nano and micro-particles. It has been reported by Zhang et al (Computational Materials Science 125 (2016) 105–109) that in order to prepare a nanoparticle with realistic atomistic structure, single silver nanoparticle with 6663 atoms is cut out from a large silver FCC supercell within a radius of 3.0 nm. In the present study, this needs to be elaborated.

Response:

Thank you for this comment. In this study, we mainly analyze the sintering diffusion of multi-scale silver in the angstrom dimension. In order to more truly simulate the diffusion bonding process of particles in the sintering process, particle pairs adopt different crystal directions, which will also lead to different atomic composition numbers in different crystal directions under particles with the same radius, so it is difficult to describe the atomic composition number of a single particle with each radius. In view of this, we modify it here to describe the number of atoms of a pair of particles. After discussion, in order to the strictness of study, we modify it here to describe the number of atoms of a pair of particles.

  1. In Fig 1 a-b, SEM images of sintered Ag surface with nanoparticles and microparticles reported by (C. Chen, K. Suganuma, Materials and Design 162 (2019) 311–321) are shown. However, in the figure caption it is written as by Kazuhiko Sugiura et. al. This needs to be clear with details of proper permissions for reuse.

Response:

Thank you for your correction for this error. In this study, we did quote the results previously reported by Prof. suganuma et. al. This error was caused by the fact that we also quoted a report of Kazuhiko Sugiura et. al in the initial draft, but later found that the same results were quoted repeatedly, so this part was deleted. It is also our negligence to fail to find this error in the revision of the original paper in time.

4.It would be interesting to explain why the authors selected multi-scale silver particles with micro and nano sizes. Why the sub-micron size was excluded for the simulation ?

Response:

Due to the computational power of molecular dynamics, we did not simulate the diffusion of micron and nan oparticles. The simulations of our research are carried out below Angstrom level. Our research is mainly to demonstrate that the mixing of large and small particles is easier to form a dense bonding layer than only large particles, so as to reveal the micro mechanism of the sintering advantage of mixed particles. Thank you again for your valuable suggestions. In the introduction part, we added a sentence: in order to further reveal the micro mechanism of the advantages of mixed scale particle sintering, we use molecular dynamics simulation to simulate the difference between the diffusion mechanism of large and small particles and that of large particles.

5.The porosity of the sintered morphology has been considered only on the surface. Considering the practical aspect, the effect of different size on porosity along the cross-section of sintered Ag paste should also be addressed as the porosity will be different on the surface and cross-section.

Response:

Thank you for this comment. We believe that the overall porosity is closer to the contact and distribution state with the actual particles, which is better than simply looking at the porosity of the cross-section.

6.Lines 45, 243, 244- particle diameter of 3.6 m, 27.6 m, 2.06 m, 0.6 m- these are serious errors.

Response:

Thank you for your correction for this error. This error is because the Arabic symbols in latex program need to use command input, which is a problem I ignored in writing. Thank you again for your correction and other places with the same errors in the article had also been corrected accordingly.

7.Line 46- feasibility of low cost nano-micron multi-scale silver particles as the potential bonding materials for sintering. It is suggested to explain how low cost can be achieved.

Response:

Thank you for this comment. In the preparation of multi-scale silver paste, we can add an appropriate amount of nano or submicron silver powder and an appropriate amount of solvent based on the relatively cheap micron silver paste to reduce the storage cost and preparation cost. In the original paper, we also add corresponding descriptions to this explanation.

  1. Line 70 - vacancy formation energy - Embedded atom method has been successfully used for predicting phase transformations, surface energy, and vacancy formation energy in metal and alloy systems. In the present context what is the relevance of prediction of vacancy formation energy?

Response:

Thank you for this comment. In the earliest manuscript, we also considered calculating the particle dislocation and vacancy energy in the densification process, but due to the current computer ability and the imperfect experimental data we can refer to, we can't continue to complete this calculation. Thank you for your comment again, and we also deleted this paragraph accordingly.

  1. How was the semi-in situ sintering monitoring conducted with the aid of SEM? This needs to be elaborated.

Response:

Thank you for your correction. We have added in-situ observation of sintering in the experimental method part of the original text. At the same time, in order to ensure the consistency of the content, we move forward the first and second paragraphs of experimental discussion 3.3.

  1. Fig.10 - How was A particle pair and B particle pair discerned from the SEM image?

Response:

Thank you for your correction. During the observation of semi in-situ sintering, resampling did not occur in the whole process, that is, the whole sintering process was sampled and observed in one area, so the corresponding position of the particle pair on the way did not change. The reader can find the corresponding particle pair by the corresponding particle pair position in Fig. 10(a) and the corresponding position in other figures.

Reviewer 2 Report

The authors presented an MD study of the sintering densification of multi-scale silver layer. This paper might be publishable after the authors address the following concerns:

  • Remove borders in the labels in Figure 2.
  • Instead of using the word chapter to identify the different parts of the manuscript, I suggest using sections instead
  • There are some typo errors such as in Figure 5, instead of rete it should be rate. In addition to this figure, the authors specified a and b as the porosity change with sintering time and change of rate of porosity, respectively but the authors probably referred to the 2 figures at each profile. The a, b, and c probably be at different particle sizes.
  • In Figure 6, what is the difference between the bullets and the lines?
  •  

Author Response

Many thank you for the positive comments and suggestions, which increased the scientific quality of this manuscript. We have revised our manuscript according to the comments, with the changes highlighted in the revised manuscript. Specific point-by-point responses to your comments are listed in the below. We believe that we have addressed the reviewer’s comments satisfactorily. We hope the revised manuscript is now suitable for publication and looking forward to hearing from you.

  1. Remove borders in the labels in Figure 2.

Response:

Thank you for correcting this error. We failed to find this problem in time in the original image, but there was a situation when latex was converted to PDF. This is an oversight in our work. We failed to carefully check all kinds of converted pictures. Thank you again for correcting this error.

  1. Instead of using the word chapter to identify the different parts of the manuscript, I suggest using sections instead.

Response:

Thank you for your valuable suggestions. We have changed the title from “Methodology of Simulation ” to “Method”, and the process of in-situ observation is added to the modeling of the original simulation.

  1. There are some typo errors such as in Figure 5, instead of rete it should be rate. In addition to this figure, the authors specified a and b as the porosity change with sintering time and change of rate of porosity, respectively but the authors probably referred to the 2 figures at each profile. The a, b, and c probably be at different particle sizes.

Response:

Thank you for correcting this error, and the relevant word error had been changed in the original paper. We are sorry that there is some ambiguity in the result expression on the picture, which is easy to make readers misunderstand. Therefore, we have changed the picture here, and thank you for feeding back this problem to us.

  1. In Figure 6, what is the difference between the bullets and the lines?

Response:

Thank you for this comment. The line is the potential energy curve of the potential energy changing with the sintering time, and the particles correspond to the morphology of the particle pair in the special reaction stage.

Reviewer 3 Report

The present manuscript describes “Molecular dynamics simulation of sintering densification of multi-scale silver layer” The authors have studied the sintering densification mechanism of silver with different mixed particle ratios Using molecular dynamics and combine with MatLab random arrangement. The authors also analyzed the neck growth mechanism of silver particle pair in the sintering process. It was found that adding small silver particles to large silver particles is beneficial to reduce porosity with the same sintering temperature and time. The x/r of the small silver particle pair is faster than 260 the large silver particle pair due to higher surface diffusion under heating at initial stage. The x/r of a small particle pair is higher than that of a large particle pair at the same temperature due to the high surface diffusion energy. Using this mechanism improves the radius of the bonding neck and shortens the sintering time by adding small silver particles to large particles. The authors reported that the multiscale silver sintering experiment, nano-silver as the medium of micron silver bonding could effectively shorten the sintering time and reduce the effective sintering temperature to improve the packaging yield. I recommend publish this good research manuscript in Materials Journal.

Author Response

Thank you very much for your recognition of the experimental data and results of this article, and I believe we can make greater breakthroughs in this field in the future.

Reviewer 4 Report

The paper “Molecular Dynamics Simulation of Sintering Densification of Multi-Scale Silver Layer” has the aim to demonstrate the feasibility of multi-scale silver sintering. This method is proved by theoretical and experimental. In this way is possible to optimize the sintering process and the preparation of silver paste for die-attach in powering electronics industry. The scientific quality of the papers is good, and the topic is interesting. The scientific method appears correct, and the research activities are well programmed.

I suggest to accept the manuscript in present form.

Author Response

(The authors gave the same response as above.)

Reviewer 5 Report

The manuscript presents a procedure to simulate the sintering densification of a silver layer. In addition, the problem is well-posed and has essential applications in the battery industry. The study is sufficiently detailed and was executed quite carefully. Therefore, I do not have any reason to doubt the manuscript results. However, I believe the manuscript will be suitable for publication after some modifications to answer some dubious points and explain the model better.

1) In line 59, it is mentioned that the molecular dynamics can be more realistic with random particle size accounting for the algorithm. What is the distribution of sizes? Is a Gaussian distribution possible? There is a most frequent size? How does the distribution of simulated particles connect with actual particle sizes?

2) In the sintering densification experiment, can one control the silver particle size? Please, discuss.

3) If I understood well, the system that will be densified has three groups of silver nanoparticles. Were other sizes simulated? How will different relative size proportions change the results?  

4) In discussing the results of Fig. 6, the authors state that the particles with lower radius can easier diffuse than the large ones. Can we speak on the different diffusion coefficients of diffusion for the particles? Can we expect any dependence of the particle diffusivity on the particle radius?

Minor changes:

1) Please change the phrase that begins in line 23: "Considering about the rapid increasing...". It is long and hard to understand.

2) Change capitalization in line 44: "A particle..."

3) Add space after punctuation in line 54: "so far.In".

4) Please change the phrase that begins in line 59: "In this paper, to simulate more authenticity...".

5) Please change capitalization in line 72.

6) Please edit the phrase that begins in line 74. A suggestion is: "The timestep unit is related to the selected atom style. In this study, we set the timestep size with the default...".

7) Please edit line 100. A suggestion is "to and achieves a more precise understanding of the densification process".

8) Please edit the phrase that begins in line 120: "During the sintering, in order...".

9) In Sec. 2.2, line 122, it is mentioned that "We set the sintering temperature to 250°C for 10 min with pressureless". Does it mean that the pressure is zero? Please, explain and edit.

10) Please, edit line 158: "corresponding driving force is following formulation[17]" to "corresponding drive force follows[17]".

11) Please change capitalization in line 160: "Where...".

12) Please edit the phrase at line 167. It is hard to understand.

13) Please edit line 193: "However, under the same simulation’s conditions..." to "However, under the same simulation conditions...".

14) Please, edit line 196: "This difference further indicated that small particles are more easily diffuse than large
particles at the same temperature" to "This difference further indicates that small particles can easier diffuse than large ones at the same temperature".

15) Please edit line 203: "However, increasing the sintering time in actual bonding will increase the packaging cost and reduce yield.". I did not understand this statement.

16) Please fix capitalization in line 206.

17) Please fix capitalization in line 217.

18) Please, edit line 240: "resulting in the also growth of the large particle pair while bonding." to "also resulting in the growth of the large particle pair while bonding."

19) Please, edit line 255: "Using molecular dynamics and combine with MatLab random arrangement, ..." to "Using molecular dynamics, combined with MatLab random arrangement, ..."

20) Please, edit the caption of Fig. 9: "MD profile of particle pair with the radius of 50 Å" to "MD profile of a particle pair with radius 50 Å".

21) the legend of figure 4 should be improved. The figure caption should mention the groups and not the sizes. The text discussed porosity in relation to groups A, B and C.

22) Figure 06 could improve the caption and mention in the caption that the lines correspond to a theoretical fit;

Author Response

Many thanks for your valuable comments leading us to improve the scientific quality of this study. We have revised our manuscript according to the comments. Please consider this paper as a publication in Materials.

  • In line 59, it is mentioned that the molecular dynamics can be more realistic with random particle size accounting for the algorithm. What is the distribution of sizes? Is a Gaussian distribution possible? There is a most frequent size? How does the distribution of simulated particles connect with actual particle sizes?

Response:

Thank you very much for your question. When simulating the random arrangement of particles, we use MatLab engineering software for calculation and modeling. The following is our specific modeling ideas. If you have any questions, please correct them in your next reply:

Before the Start step, we need to establish a simulation box and the radius of the circle in the box, and the ratio of circles with different radii. Then, we let the center coordinate y = r to model the first layer of circles. During modeling the first layer, modeling needs to follow the ratio of circle radius which we had set in before. After the first layer of modeling is completed, we start to follow the following flow chart to model the remaining particles until the simulation box is filled.

(Pictures cannot be imported into the page. We have integrated the problem and revised draft into PDF)

Because this algorithm involves project confidentiality agreements, we are sorry that we cannot clearly describe the calculation process.

In applications, the particles we see the longest include nanoscale (≤50 nm ), submicron scale (≥100 nm and<1 μm), and micron-scale (≤5 μm). Due to the limited of molecular dynamics and the computing power of the computer, we cannot simulate micron silver particles and nanosilver particles. All the simulations were carried out under Ångstrom (1 Å = 0.1 nm). In this study, we mainly want to verify the sintering mechanism of multi-scale silver through the micro-level, to study the sintering densification mechanism of multi-scale silver from the micro-level.

When selecting particles, we mainly consider two factors for radius selection: The computing power of the computer and the minimum undistorted radius in particle modeling. In molecular dynamics modeling, particles are stacked by atoms according to rules. If the selected minimum radius is too small, the model will be distorted and the simulation results will not be good. At the same time, the modeling effect of too large particle radius is much better, but it will take more time and computing power in the calculation. Because the maximum simulation time of LAMMPS in molecular dynamics is only 100ms, we chose the minimum undistorted radius for modeling.

  • In the sintering densification experiment, can one control the silver particle size? Please, discuss.

Response:

Thank you for your question which makes our study more scientific. In the current technology of preparing silver powder, the particle size and average particle size of silver particles prepared by the chemical reduction method have been well controlled. For example, in the preparation of the reaction, we need to accurately control the selection of reducing agent, water bath heating temperature, stirring rate and drop acceleration of organic matter, so as to prepare the required homogeneous particles.

  • If I understood well, the system that will be densified has three groups of silver nanoparticles. Were other sizes simulated? How will different relative size proportions change the results? 

Response:

Thank you for your question. Due to the limited molecular dynamics and the computing power of the computer, we cannot simulate micron silver particles and nanosilver particles. All the simulations were carried out under Ångstrom (1 Å = 0.1 nm). In this study, we mainly want to verify the sintering mechanism of multi-scale silver through the micro-level, to study the sintering densification mechanism of multi-scale silver from the micro-level. In the early modeling, we also selected several groups of particle combinations for relevant simulation experiments, such as 10 Å,20 Å,50 Å; and 20 Å,30 Å,60 Å. However, considering the computer computing power and reasonable proportional size difference, we finally selected 10 Å,30 Å,50 Å. Here, in terms of particle selection, the smaller the radius of small particles is, the better its bonding ability will be. However, we should also consider the impact of the real problem of the model on the simulation results.

  • In discussing the results of Fig. 6, the authors state that the particles with lower radius can easier diffuse than the large ones. Can we speak on the different diffusion coefficients of diffusion for the particles? Can we expect any dependence of the particle diffusivity on the particle radius?

Response:

Thank you very much for your question. The relationship between particle radius and sintering neck is as follows[1]:

(3)

Where t is the sintering time, n and m are constants dependent on the specific transport mechanism, and B is a term made of material and geometric constants. Which is also explained in line 198 of the original paper. When the particle radius decreases, the surface energy of the whole silver powder will also increase. This increased surface energy provides sintering driving force in the sintering process[2]:

Where σ is the sintering driving force, γ is the material surface energy, and R1 and R2 are the radii of two particles that diffuse each other. Which is also explained in line 159 of the original paper.

[1] Zhang Y, Wu L, Guo X, Jung YG, Zhang J. Molecular dynamics simulation of electrical resistivity in sintering process of nanoparticle silver inks. Computational Materials Science. 2016;125:105-9.

[2]Alpha. https://www.psma.com/sites/default/files/uploads/files/Silver Sintering – Myths %26 Physics.pdf

Minor changes:

Response:

Thank you for your valuable comments on the structure and language of this article. Thank you for correcting some omissions in our writing. At the same time, we have revised the relevant questions you raised in the original text. The specific amendments have been indicated in the corresponding positions in the revised draft. With regard to your Q. 9, I'm sorry that the use of "pressureless" in the original text has misled your reading. We also delete this word in the original text.

The traditional silver sintering process is divided into pressure assisted sintering and pressureless sintering shown in the fig.1.  Here, we use the word "pressureless" to further explain that our experiment does not add any pressure, that is, what you call "zero pressure", while ignoring the common sense that only pressureless sintering can be carried out during SEM semi-in-situ sintering. Once again, thank you for your valuable suggestions on the integrity and scientificity of this article, which enable us to do better

Fig.1 Schematic diagram of different sintering processes

Minor changes:

1) Please change the phrase that begins in line 23: "Considering about the rapid increasing...". It is long and hard to understand.

2) Change capitalization in line 44: "A particle..."

3) Add space after punctuation in line 54: "so far.In".

4) Please change the phrase that begins in line 59: "In this paper, to simulate more authenticity...".

5) Please change capitalization in line 72.

6) Please edit the phrase that begins in line 74. A suggestion is: "The timestep unit is related to the selected atom style. In this study, we set the timestep size with the default...".

7) Please edit line 100. A suggestion is "to and achieves a more precise understanding of the densification process".

8) Please edit the phrase that begins in line 120: "During the sintering, in order...".

9) In Sec. 2.2, line 122, it is mentioned that "We set the sintering temperature to 250°C for 10 min with pressureless". Does it mean that the pressure is zero? Please, explain and edit.

10) Please, edit line 158: "corresponding driving force is following formulation[17]" to "corresponding drive force follows[17]".

11) Please change capitalization in line 160: "Where...".

12) Please edit the phrase at line 167. It is hard to understand.

13) Please edit line 193: "However, under the same simulation’s conditions..." to "However, under the same simulation conditions...".

14) Please, edit line 196: "This difference further indicated that small particles are more easily diffuse than large particles at the same temperature" to "This difference further indicates that small particles can easier diffuse than large ones at the same temperature".

15) Please edit line 203: "However, increasing the sintering time in actual bonding will increase the packaging cost and reduce yield.". I did not understand this statement.

16) Please fix capitalization in line 206.

17) Please fix capitalization in line 217.

18) Please, edit line 240: "resulting in the also growth of the large particle pair while bonding." to "also resulting in the growth of the large particle pair while bonding."

19) Please, edit line 255: "Using molecular dynamics and combine with MatLab random arrangement, ..." to "Using molecular dynamics, combined with MatLab random arrangement, ..."

20) Please, edit the caption of Fig. 9: "MD profile of particle pair with the radius of 50 Å" to "MD profile of a particle pair with radius 50 Å".

21) the legend of figure 4 should be improved. The figure caption should mention the groups and not the sizes. The text discussed porosity in relation to groups A, B and C.

22) Figure 06 could improve the caption and mention in the caption that the lines correspond to a theoretical fit;

Round 2

Reviewer 1 Report

The authors have appropriately modified the manuscript and hence I would like to recommend the publication of this article.

However, still the units of particle size and distance between particles is provided as in metres (m) which should be corrected to micrometres (µm) in the final version.

Author Response

However, still the units of particle size and distance between particles is provided as in metres (m) which should be corrected to micrometres (µm) in the final version.

Response:

Many thank you for pointed out this error again. We are sorry to ignore this problem again in the first modification. This time, after careful examination, we have confirmed that we have completely corrected this error. Thank you again for your valuable comments on the scientific rationality and structural integrity of this article
